# Left-handed DNA for efficient highly multiplexed imaging at single-protein resolution

Eduard M. Unterauer[1,2,8], Eva-Maria Schentarra [1,2,8], Isabelle Pachmayr [2,3,8], Taisha Tashrin[2,8], Jisoo Kwon[1], Sebastian Strauss [2], Kristina Jevdokimenko[4], Rafal Kowalewski [1,2], Felipe Opazo [4,5,6], Eugenio F. Fornasiero [4,7], Luciano A. Masullo [2] ✉ & Ralf Jungmann [1,2] ✉

Multiplexed super-resolution microscopy enables spatial proteomics at single-protein resolution, but current methods often depend on secondary labels, complicating implementation and limiting throughput. We introduce a streamlined approach that combines speed-optimized DNA-PAINT sequences with their mirror-image analogs (left-handed DNA), enabling rapid and efficient 12-plex imaging. Validated on synthetic and cellular benchmarks, our method maps dense neuronal interactomes in 3D with 15 nm spatial resolution across a $200 \times 200\,\mu m^2$ field of view.

Understanding the detailed molecular architecture of cells is of fundamental interest in the life sciences, as any macroscopic biological phenomenon has its origin in specific molecular interactions and arrangements on the nanoscale. Many approaches, such as imaging mass cytometry (IMC)[1], multiplexed ion beam imaging (MIBI)[2] or co-detection by indexing CODEX[3], have had a substantial impact on revealing this complex heterogeneity on a subcellular level. However, their spatial resolution is limited to about 200 nm, precluding studies of protein interactions at the molecular scale.

Super-resolution microscopy, on the other hand, can achieve molecular spatial resolution, but multiplexing is generally limited to a few protein species since fluorescent proteins and organic dyes have a limited number of distinguishable emission wavelengths. Methods like Exchange-PAINT[4], DNA-PRISM[5], and maS³TORM[6] can extend the multiplexing to above 10 "channels" but are restricted to long acquisition times. The recently developed FLASH-PAINT[7] and SUM-PAINT[8] methods overcome this limitation by combining speed-optimized DNA-PAINT readout strands[9] with primary DNA barcoding and secondary labeling probes, enabling theoretically unlimited multiplexing with up to a 100-fold increased speed and single-protein resolution. However,

both methods require somewhat intricate DNA sequence design and complex experimental optimization. These methods rely on either stable or transient secondary DNA barcodes, requiring additional hybridization and signal extinction steps, or involving complex strand displacement dynamics. Such intricacies pose significant barriers for routine adoption, particularly by non-expert users.

On the other hand, left-handed DNA-PAINT[10] (L-DNA-PAINT) was introduced to enhance imaging in nuclear environments by minimizing non-specific interactions and thereby reducing background signal. Although L-DNA-PAINT achieves comparable sequence specificity and spatial resolution to conventional (right-handed) DNA-PAINT, its potential for high-throughput and multiplexed imaging has remained unexplored.

Here, we introduce speed-optimized left-handed DNA-PAINT as a tool for efficient multiplexing, while featuring a straightforward and robust implementation. To characterize the sequence performance, we conducted sub-5-nm resolution imaging of DNA origami nanostructures and cellular benchmarking structures and evaluated the sequence kinetics. Our results indicate that left-handed DNA-PAINT performs similarly to the right-handed analogs both in terms of spatial

¹Faculty of Physics and Center for Nanoscience, Ludwig Maximilian University, Munich, Germany. ²Max Planck Institute of Biochemistry, Martinsried, Germany. ³Department of Chemistry and Biochemistry, Ludwig Maximilian University, Munich, Germany. ⁴Institute of Neuro- and Sensory Physiology, University Medical Center Göttingen, Göttingen, Germany. ⁵Center for Biostructural Imaging of Neurodegeneration (BIN), University Medical Center Göttingen, Göttingen, Germany. ⁶NanoTag Biotechnologies GmbH, Göttingen, Germany. ⁷Department of Life Sciences, University of Trieste, Trieste, Italy. ⁸These authors contributed equally: Eduard M. Unterauer, Eva-Maria Schentarra, Isabelle Pachmayr, and Taisha Tashrin. ✉e-mail: masullo@biochem.mpg.de; jungmann@biochem.mpg.de

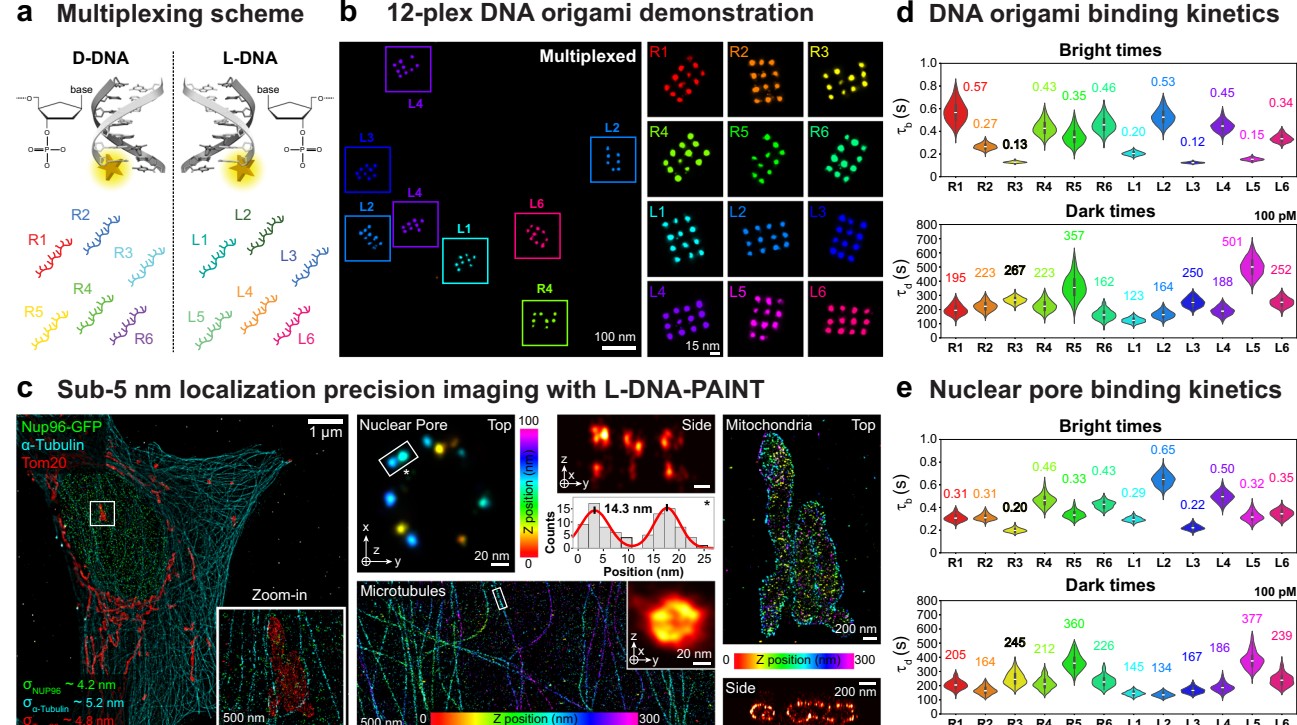

**Fig. 1 | Left-handed speed-optimized DNA-PAINT optimization. a** Illustration of 12-plex multiplexing capability with six right-handed (R1, R2, R3, R4, R5, R6) and six left-handed (L1, L2, L3, L4, L5, L6) sequences. **b** 12-plex DNA origami demonstration. Left shows an exemplary field-of-view with respective left-handed and right-handed 15 nm DNA origami grids highlighted. Right: Exemplary 15 nm DNA origami grid for each sequence. This data belongs to one dataset. **c** Sub-5 nm localization precision imaging in 3D of the three super-resolution benchmarking targets, nuclear pores (Nup96 - green - L2), tubulin filaments (α-tubulin - cyan - L1), and mitochondria (Tom20 - red - L4). Left: Overview of a single cell and a zoom-in with all proteins. Middle top: individual nuclear pore with single Nup96 copies, spaced ~14.3 nm apart (± 2 nm), resolved and a side view of the nuclear pore with the two rings spaced ~ 50 nm apart. Middle bottom: Exemplary 3D colored section of tubulin filaments with a zoom in into a cross-section of an individual tubulin filament of ~ 30 nm diameter. Right: Mitochondria with a cross-sectional side view, z-colored. **d** Kinetics characterization on DNA origami. The top violin plot shows the bright time (s) evaluation, while the bottom shows the dark time evaluation of all 12 sequences. Values are normalized to 100 pM imager concentration (**e**) Kinetics characterization in nuclear pores. The top violin plot shows the bright time (s) evaluation, while the bottom shows the dark time evaluation of all 12 sequences. Values are normalized to 100 pM imager concentration. In d and e, black lines indicate the minimum and maximum of the underlying data, black boxes indicate the two central quartiles, and the white line shows the median of the distribution. For details see Supplementary Figs. 2 and 3.

resolution and kinetics. Combining the sequence space of the right-handed and left-handed probes, we demonstrate a 13-plex neuronal atlas experiment, revealing the nano-organization of neuronal cytoskeleton, organelle markers and synapses in a $200 \times 200\,\mu m^2$ field of view (FOV). This approach serves as a simple and straightforward implementation of highly multiplexed DNA-PAINT imaging and enables subcellular spatial proteomics across scales.

## Results

### Characterization of speed-optimized left-handed DNA-PAINT

By leveraging the concept of right-handed speed-optimized sequences[9] (R1-R6), we designed concatenated and hair-pin-free analogs synthesized from mirror base pairs, which we call (L1–L6) (Supplementary Table 1–3). This allowed us to directly combine right- and left-handed DNA sequences to obtain a library for speed-optimized multiplexing of up to 12 targets (Fig. 1a). This new library can be directly used in a straightforward Exchange-PAINT workflow without the need for secondary DNA labels[7,8].

To demonstrate the capabilities of this 12-channel multiplexing approach, we designed a set of 12 orthogonal DNA origami, extended with orthogonal right- or left-handed DNA binding sites spaced 15 nm apart. We then imaged the sample with 12 rounds using right- and left-handed Exchange-PAINT probes. The resulting 12-plex image is shown in Fig. 1b, with a zoom-in into individual DNA origami on the right. We confirmed that there is no crosstalk between sequences,

and single sites spaced 15 nm apart were clearly resolved. To further benchmark the achievable resolution, we imaged 5-nm DNA origami. We resolved these structures for the L1 sequence, clearly separating 5-nm sites with a localization precision of ~1.4 nm (see Supplementary Fig. 1).

After validation in synthetic nanostructures, we evaluated the performance of the L-DNA strands in cellular environments. We chose three different subcellular targets in U2OS cells: The nuclear pore protein Nup96 coupled to GFP, the outer mitochondrial membrane protein Tom20, and the tubulin subunit alpha-Tubulin, which have all previously been established as super-resolution benchmarking structures[11]. We achieve sub-5 nm localization precision for all three benchmarking targets in 3D (see Fig. 1c). This enables us to distinguish individual Nup96 copies spaced ~13 nm apart, resolve individual Tom20 proteins in the mitochondrial outer membrane, and visualize individual tubulin filaments (~30 nm in diameter due to the linkage error introduced by our labeling strategy).

Next, to evaluate the binding kinetics of the extended speed-optimized L-DNA sequences compared to their right-handed counterparts, we analyzed DNA origami in vitro and nuclear pore complexes (NPCs) in situ. The respective "bright" and "dark" times were extracted and plotted in a violin plot (see Fig. 1d, e). Left-handed and right-handed sequences show speed-optimized kinetics comparable to previously reported values[9] (Supplementary Figs. 2, 3 and Supplementary Tables 4 and 5).

However, there are certain variations in binding kinetics between DNA origami and NPC experiments as well as between left- and right-handed sequences. The differences in the former case could potentially be attributed to differences in experimental settings: DNA origami strand extensions are hybrid sequences transitioning from right-handed to left-handed motifs in a single strand, which possibly alters the base-stacking interactions in the vicinity of the transition. The differences in the latter case, specifically the bright times of R2 and L2 differ in both the DNA origami and NPC contexts, a phenomenon for which we currently lack an explanation.

### Revealing a 13-plex neuronal atlas at single protein resolution

After benchmarking the left-handed speed sequences, we applied them in combination with their right-handed counterparts to visualize a detailed map of a complex and dense neuronal interactome. With the improved throughput of our right-handed and left-handed speed-optimized library, we were able to image a 13-plex (12-plex Exchange-PAINT plus Actin imaging using Lifeact[12], experimental conditions in Supplementary Tables 6, 7), $200 \times 200\ \mu m^2$ 3D super-resolution atlas in as little as 10 h (2.5 h per $100 \times 100\ \mu m^2$ FOV) at ~12 nm spatial resolution (~5 nm localization precision, see Supplementary Table 8). A comparable experiment with conventional Exchange-PAINT[4] would have taken more than a month to complete (see "Methods" and Supplementary Table 9). This multiscale spatial proteomics experiment enables the visualization of an interaction map spanning multiple neurons and their synapses, down to individual proteins across 13 distinct targets−effectively bridging the gap between inter-neuronal networks and the molecular architecture of single synapses.

Figure 2 shows an overview of the entire $200 \times 200\ \mu m^2$ field-of-view (FOV) next to all 13 individual protein targets with their respective localization precisions (Fig. 2b and Supplementary Figs. 4–7). Investigating this neuronal atlas allows us to detect excitatory and inhibitory synapses with their scaffold proteins bassoon, PSD95 (excitatory), and Gephyrin (inhibitory), as well as their respective neurotransmitter transporter proteins VGlut1 (excitatory) and VGAT (inhibitory) at single-protein resolution (Fig. 2c). In the super-resolved neuronal atlas, we also found synapses of the recently reported mixed-type synapse[8], with a Gephyrin postsynaptic scaffold and a VGlut1 neurotransmitter transporter. Furthermore, we generated single-protein-resolved spatial maps of key organelles, including mitochondria, peroxisomes, and the Golgi apparatus. These maps allowed us to uncover potential contact sites between peroxisomes (Pmp70) and the Golgi (Golga5) (Fig. 2d), which have recently been proposed as rare cellular events[13]. Such contact sites may represent membrane contact sites, potentially enabling direct lipid or protein exchange and suggesting previously underappreciated crosstalk between these organelles. These 3D renderings of cytoskeletal and organelle protein distributions are shown in Fig. 2e–g, highlighting βII-spectrin ring structures (Fig. 2e), neuro-filament fibers (Fig. 2f), and mitochondria traversing axons and dendrites with distinct morphologies (Fig. 2g).

## Discussion

In conclusion, we have introduced an improved high-throughput super-resolution multiplexing approach, combining speed-optimized right-handed and left-handed DNA-PAINT sequences. This allows for the fastest-to-date DNA-PAINT multiplexing for up to 12 targets. We note that our approach can be combined with approaches like peptide-PAINT[14] (in this case, Lifeact[12] for a 13-plex image), further expanding the multiplexing.

By leveraging the expanded sequence space, accelerated imaging kinetics, and straightforward experimental design, we achieved highly multiplexed cellular interaction maps within just 10 h−capturing a $200 \times 200\ \mu m^2$ field of view while resolving individual proteins spaced merely 10 nm apart. This spatial coverage across more than four orders of magnitude enables simultaneous investigation of molecular and network-level features. Our approach compresses what would traditionally require up to a month of imaging into a single working day, opening new avenues for high-throughput studies of synaptic architecture and plasticity across complex multi-neuron systems.

We were able to map the interaction patterns between neighboring neurons, trace back axonal connections, and resolve individual molecular compositions in the vesicle pools of each engaged synapse. The simultaneous visualization of intricate neuronal networks and the molecular composition of their interaction points represents a powerful approach to studying detailed cellular structures in both health and disease. By enabling the visual comparative analysis of numerous individual neurons and their molecular contents, this method allows for the identification of structural and molecular alterations in a high-throughput manner. Such insights could advance our understanding of disease mechanisms and enable the development of novel and more precise therapeutic approaches. Finally, we note that this technique is not limited to exploring neuronal targets: our approach can be extended to any biological system for which primary affinity reagents are available, enabling spatial proteomics at the single-protein level across scales for any cellular system.

## Methods

### Materials

D-DNA oligonucleotides, including C3-azide, Cy3B, and biotin-modified oligonucleotides, were purchased from Metabion and Integrated DNA Technologies. L-DNA imagers (Cy3B conjugated) and azide-modified strands, as well as chimeric D-DNA and R-DNA secondary sequences, were custom-ordered from Biomers. M13mp18 scaffold was obtained from Tilibit. Magnesium 1 M (cat: AM950G), sodium chloride 5 M (cat: AM9759), ultrapure water (cat: 10977-035), Tris 1 M (cat: AM9855G), EDTA 0.5 M (cat: AM9260G), 10×PBS (cat: 70011051), 1xPBS (cat: 20012-019), Neutravidin (cat: 3100), McCoy's 5 A media (cat: 16600082), 0.05 % Trypsin−EDTA (cat: 25300-054), FBS (cat:10500-064) and glass slides (cat:10756991) were purchased from Thermo Fisher Scientific. Triton X-100 (cat: 6683.1) was purchased from Carl Roth. Paraformaldehyde (cat: 15710) was obtained from Electron Microscopy Sciences. Bovine serum albumin (cat: A4503-10G), BSA-Biotin (cat: A8549), Tween 20 (cat: P9416-50ML), glycerol (cat: 65516-500 ml), ethylencarbonate (cat: E26258), methanol (cat: 32213-2.5 L), protocatechuate 3,4-dioxygenase pseudomonas (PCD) (cat: P8279), 3,4-dihydroxybenzoic acid (PCA) (cat: 37580-25G-F) and (+−)-6-hydroxy-2,5,7,8- tetra-methylchromane-2-carboxylic acid (Trolox) (cat: 238813-5 G) were ordered from Sigma Aldrich. Coverslips (cat: 0107032) were purchased from Marienfeld. Double-sided tape (cat: 665D) was ordered from Scotch. Two-component silica twinsil speed 22 (cat. 1300 1002) was purchased from Picodent. 90 nm diameter Gold Nanoparticles (cat: G-90-100) were ordered from cytodiagnostics. Dextran sulfate 50% solution (cat: E516-100ML) was purchased from VWR.

### Buffers

The following buffers were used for sample preparation and imaging:

- Buffer C + : 1× PBS, 500 mM NaCl and 0.05% Tween-20
- Buffer C: 1× PBS, 500 mM NaCl
- Antibody Incubation buffer: 1× PBS, 1 mM EDTA, 0.02% Tween-20, 0.05% NaN₃, 2% BSA, and 0.05 mg/ml sheared salmon sperm DNA
- Blocking Buffer: 1x PBS, 3% BSA, 0.25%Triton X-100 and 0.05 mg/ml sheared salmon sperm DNA
- Buffer A + : 10 mM Tris pH 8, 100 mM NaCl and 0.05% Tween-20
- Buffer B + : 10 mM MgCl2, 5 mM Tris-HCl pH 8, 1 mM EDTA and 0.05% Tween-20, pH 8
- Buffer B: 10 mM MgCl2, 5 mM Tris-HCl pH 8 and 1 mM EDTA, pH 8
- Optimized hybridization buffer: 10% Dextran Sulfate, 10% Ethylencarbonate, 4xSSC and 0.4 % Tween-20

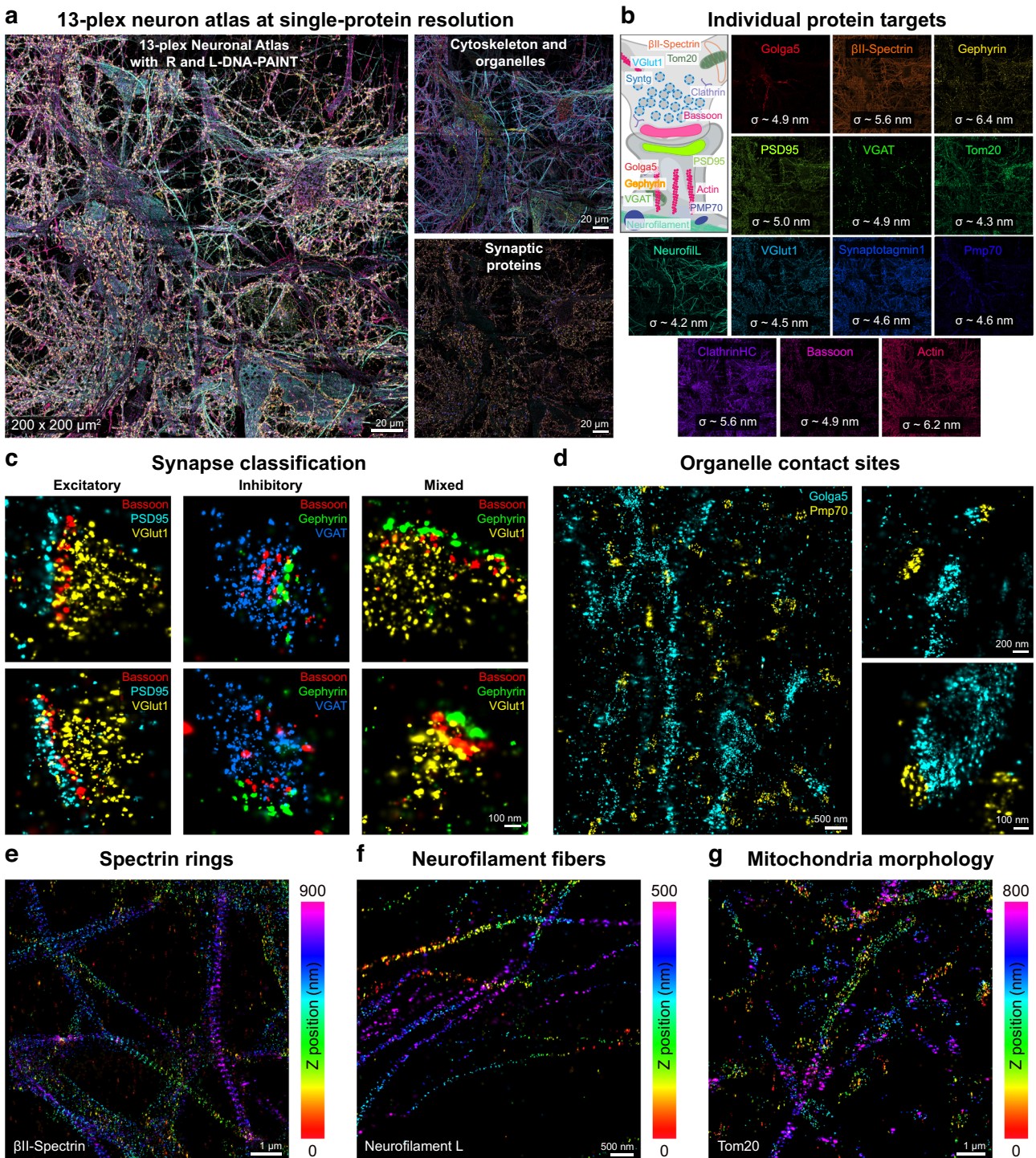

**Fig. 2 | 13-plex neuronal atlas at single-protein resolution. a** 200 × 200 µm² 3D neuronal atlas at single-protein resolution. Left: The entire field of view with all 13 protein targets overlayed. Right top: cytoskeleton (βII-Spectrin, Neurofilament light-chain (NeurofilL), Actin) and organelle (Golga5, Tom20, Pmp70) proteins. Right bottom: Synaptic proteins (Gephyrin, PSD95, VGAT, VGlut1, Synaptotagmin1, Bassoon). **b** All 13 individual protein targets shown in an illustration next to their single-channel image in (**a**), with their respective localization precision (on average, ~5 nm). **c** Synapse classification, showing selected excitatory (Bassoon, PSD95, VGlut1), inhibitory (Bassoon, Gephyrin, VGAT), and mixed (Bassoon, Gephyrin, VGlut1) synapses with single molecules resolved. **d** Organelle contact site mapping between peroxisomes (Pmp70) and the Golgi apparatus (Golga5). Left: Exemplary field of view with Golga5 rendered in cyan and Pmp70 in yellow. Right: A selection of possible contact sites in an independent zoom in. **e** Spectrin (βII-Spectrin) distribution in an exemplary field of view, z-colored, with well-resolved ring-like structures. **f** Neurofilament (NeurofilL) distribution in an exemplary field of view, z-colored, with single fibers resolved. **g** Mitochondria (Tom20) distribution in an exemplary field of view, z-colored, with elongated and spherical morphologies. This data belongs to one dataset.

## Trolox, PCA, and PCD

100× Trolox: 100 mg Trolox was added to 430 μl 100 % Methanol, 345 μl 1 M NaOH, and 3.2 ml $H_2O$. 40× PCA: 154 mg PCA, 10 ml water and NaOH were mixed, and pH was adjusted to 9.0. 100× PCD: 9.3 mg PCD, 13.3 ml of buffer (100 mM Tris-HCl pH 8, 50 mM KCl, 1 mM EDTA, 50 % Glycerol).

## Microscopy setup

Fluorescence imaging was carried out on an inverted microscope (Nikon Instruments, Eclipse Ti2) with the Perfect Focus System, applying an objective-type TIRF configuration equipped with an oil-immersion objective (Nikon Instruments, Apo SR TIRF×100, NA 1.49, Oil). A 560-nm laser (MPB Communications, 1 W) was used for excitation and coupled into the microscope via a Nikon manual TIRF module. The laser beam was passed through a cleanup filter (Chroma Technology, ZET561/10) and coupled into the microscope objective using a beam splitter (Chroma Technology, ZT561rdc). Fluorescence was spectrally filtered with an emission filter (Chroma Technology, ET600/50 m, and ET575lp) and imaged on an sCMOS camera (Hamamatsu Fusion BT) without further magnification, resulting in an effective pixel size of 130 nm after 2 × 2 binning. TIR illumination was used for all measurements. The central 1152 × 1152 pixels (576 × 576 after binning) of the camera were used as the region of interest. The scan mode of the camera was set to "ultra quiet scan" (readout noise = 0.7 e- r.m.s., 80 μs readout time per line). Raw microscopy data were acquired using μManager (Version 2.0.1)[15]. 3D super-resolution reconstruction by astigmatism was achieved by introducing a cylindrical lens in the detection pathway[16].

## Animals

Wild-type Wistar rat pregnant mothers or pups (*Rattus norvegicus*) and adult mice (*Mus musculus*) were obtained from the University Medical Center Göttingen and were handled according to the specifications of the University of Göttingen and of the local authority, the State of Lower Saxony (Landesamt für Verbraucherschutz, LAVES, Braunschweig, Germany). Animal experiments were approved by the local authority, the Lower Saxony State Office for Consumer Protection and Food Safety (Niedersächsisches Landesamt für Verbraucherschutz und Lebensmittelsicherheit).

## Primary cell culture

Primary hippocampal neuron cultures from embryonic day 18 (E18) Wistar rat embryos were prepared similarly to previous work[17]. In brief, upon dissection, neurons were grown on poly-L-lysine coated coverslips (1 mg/ml) over an astrocyte feeder layer and were kept in an N2-supplemented serum-free medium. Glial cells were prepared from P2 Wistar rat pups and seeded in 12-well plates at a density of 10.000 cells per well, three days prior to the preparation. In the next step, hippocampal neurons were seeded onto 18 mm Ø coverslips at a density of 60.000 cells per coverslip. Paraffin dots on the 18 mm Ø coverslips were acting as a spacer between the neurons and glial cells. The cell culture medium was exchanged with fresh medium twice a week. Following this protocol, the neurons developed proper polarity, generated intricate axonal and dendritic networks, and established multiple functional synaptic connections with each other[18].

## Nanobody-DNA conjugation via single cysteine

Nanobodies against GFP, rabbit, and mouse kLC were purchased from Nanotag with a single ectopic cysteine at the C-terminus for site-specific and quantitative conjugation. The conjugation to DNA-PAINT docking sites (see Supplementary Table 2) was performed as described previously[19]. First, the buffer was exchanged to 1× PBS + 5 mM EDTA, pH 7.0, using Amicon centrifugal filters (10k MWCO), and free cysteines were reacted with a 20-fold molar excess of bifunctional maleimide-DBCO linker (Sigma Aldrich, cat: 760668) for 2–3 h on ice.

The unreacted linker was removed by buffer exchange to PBS using Amicon centrifugal filters. Azide-functionalized DNA was added with 3–5 molar excess to the DBCO-nanobody and reacted overnight at 4 °C. Unconjugated nanobody and free azide-DNA were removed by anion exchange using an ÄKTA Pure liquid chromatography system equipped with a Resource Q 1 ml column. Nanobody-DNA concentration was adjusted to 5 μM (in 1xPBS, 50% glycerol, 0.05% $NaN_3$) and stored at −20 °C. An exemplary anion exchange chromatogram of DNA, unconjugated DNA and DNA-conjugated GFP nanobody can be seen in Supplementary Fig. 8.

## 12-plex DNA origami experiment and additional DNA origami kinetics measurements

Origami sample preparation was done in a 6-channel μ-slide (Ibidi Cat.: 80607). First 50 μl of biotin-labeled bovine albumin (1 mg/ml, dissolved in buffer A + ) was flushed into the chamber and incubated for 3 min. The chamber was subsequently washed with 1 ml of buffer A+ followed by incubation with 200 μl of neutravidin (0.5 mg/ml, dissolved in buffer A + ) for 3 min. Afterward, the chamber was washed again with 1 ml of A+ and 1 ml of B+ buffer and incubated with biotin-labeled DNA origami (~100 pM per species, 1.2 nM in total in buffer B + ) for 3 min. Subsequently, the sample was washed with 1 ml B+ buffer and 1 ml 2xSSC buffer. DNA origami were hybridized to secondary labels carrying right-handed complements to DNA origami staple strands and a left-handed or a right-handed DNA-PAINT readout site Supplementary Tables 2, 3 (secondary-labels were chosen because of cost efficiency) in optimized hybridization buffer at a concentration of 100 nM per strand for 15 min. Finally, the chamber was washed 5x with 1 ml 2xSSC buffer and 1x with 1 ml of C+ buffer and subsequently incubated with 1 ml imager solution (Buffer C+ plus the respective imager concentration, see Supplementary Table 10), and 15.000 frames were taken with 75 ms exposure time at a laser power of 50 mW at the sample (in TIRF mode). After acquisition, the sample was washed with 2 ml buffer B, after the next imager solution was applied until all 12 targets were acquired.

## 5 nm DNA origami experiment

Origami sample preparation was done similarly to the 12-plex experiment, with the exception of incubating ~200 pM 15 nm drift correction DNA origami structures and ~100 pM 5 nm MPI-Logo DNA origami for 3 min in buffer B + . Subsequently, the sample was washed with 1 ml B+ buffer and 1 ml 2xSSC buffer. 100 nM of the respective secondary-label strand (BC 30 to L6) was incubated in an optimized hybridization buffer for 15 min. Secondary labels were used here as well, out of cost efficiency. Afterward, the sample was washed 5 times with 2xSSC buffer and once with buffer C+ and incubated with 1 ml 50 pM L6-Cy3B imager solution. The sample was then imaged in TIRF mode with 60 mW laser power at the sample for 100,000 frames at 100 ms exposure time.

## 3-plex imaging in U2OS cells

U2OS Nup96-GFP cells (a gift from the Ries and Ellenberg laboratories) were cultured in McCoy's 5 A medium (Thermo Fisher Scientific, 16600082) supplemented with 10% FBS. Approximately 30 K cells per $cm^2$ were seeded the day before fixation. Cells were fixed with 4% PFA and 0.1% GA in PBS for 15 min at RT and washed with 3x PBS. Then, cells were permeabilized with 0.2% Triton for 30 min and blocked with antibody incubation buffer for 45 min at room temperature. Primary antibodies rabbit anti-Tom20 (Abcam) and mouse-anti-α-Tubulin (Sigma) were diluted in antibody incubation buffer 1:200 and incubated overnight at 4 °C. See Supplementary Table 6 for details about the antibodies.

The next day, the primary antibody solution was aspirated, and the sample was washed 3x with PBS. Then, the (secondary) single domain antibodies (sdAbs) anti-mouse kappa light chain sdAb-5xL1,

anti-rabbit sdAb-7xL4, and anti-GFP-sdAb-5xL2 (Nanotag 1H1) were diluted to 25 nM each in antibody incubation buffer and incubated for 1.5 h at RT. Cells were washed 2x with 1xPBS + 0.02%Tween, followed by 2x PBS. The stained sample was post-fixed with 4% PFA in PBS for 10 min, followed by washing 3x with PBS. The sample was quenched with 0.2 M NH4Cl in PBS for 5 min, followed by washing 3x with PBS. Then, cells were incubated with gold NPs, diluted 1:3 in PBS for 7 min, followed by 2 washes with PBS.

For imaging, the focal plane was set using the Nup96-GFP signal. Cells were imaged in HILO[20] at a laser power of 18 mW. 100 pM L2-Cy3B imager solution was applied to the sample, and 60.000 frames were acquired for Nup96-GFP. Afterwards, the sample was washed 4x with PBS, and 25 pM L1-Cy3B imager solution was applied for imaging of tubulin for 120.000 frames. Finally, after washing with 4xPBS, 50 pM L4-Cy3B containing imager solution was flushed onto the sample, and 60.000 frames were acquired to image Tom20.

## Nuclear pore kinetic measurements

U2OS Nup96-GFP cells (a gift from the Ries and Ellenberg laboratories) were cultured in McCoy's 5 A medium (Thermo Fisher Scientific, 16600082) supplemented with 10% FBS. Approximately 10 K cells per cm$^2$ were seeded the day before fixation. Cells were fixed with 4% PFA in PBS for 15 min at RT and washed with 3x PBS. Then, the fixed cells were permeabilized with 0.2% Triton-X in PBS for 30 min followed by 3x PBS wash. Afterward, the cells were blocked with antibody incubation buffer for 1 h at room temperature. The samples were then washed with 3x PBS and incubated with 1:200 anti-GFP-sdAb (for all respective targets individually, R1-R6 and L1-L6) in antibody incubation buffer overnight at 4 °C. The next day, the samples were washed 3x with PBS and postfixed with 4% PFA in PBS for 10 min at room temperature. After 3x PBS washed, the samples were incubated with 1:3 diluted gold nanoparticles in PBS for 5 min, followed by 5x PBS washing. Prior to imaging, the samples were incubated with 100 pM of the respective imaging solution according to Supplementary Table 10 and imaged for 40.000 frames at 20 mW laser power.

## Single-molecule localization analysis

Raw fluorescence data were reconstructed using the Picasso software package[21] (the latest version is available at https://github.com/jungmannlab/picasso). Drift correction was performed using the AIM algorithm[22] with gold nanoparticles as fiducials for all experiments. Alignment between different channels was done through cross-correlation of the fiducial gold nanoparticles using Picasso.

## DNA origami binding kinetics analysis

For binding kinetics calculation, individual acquisition rounds were clustered using single-molecule clustering in Picasso[23] with a cluster radius of 6.5 nm, a minimal localization number of 5, and basic frame analysis to enable single binding site detection. Binding kinetics were then calculated on entire DNA origami structures by extracting the mean bright and dark time of > 900 individual DNA Origami (picked with the Picasso *pick similar* tool) with a radius of 1 pixel (130 nm) with the Picasso software package[21]. The respective mean dark times were then normalized by the imager concentration of the measurement (see Supplementary Table 10) and the average binding sites detected by single-molecule clustering. The values were then combined in Supplementary Table 4.

## Nuclear pore binding kinetics analysis

For binding kinetics calculation, 500 nuclear pores were picked with the Picasso[23] pick tool at a radius of 2 pixels (260 nm) for each imager sequence (R1-R6 and L1-L6). The binding kinetics were then calculated on the entire nuclear pore structures by extracting the mean bright and dark times of all nuclear pores. Finally, the respective values were normalized by the expected number of

NUP96 copies (16 at ~50% labeling efficiency[24]) representing single binding site kinetics. The values were then combined in Supplementary Table 5.

## 13-plex neuron imaging

Primary rat hippocampal neurons were fixed with preheated (37 °C) 4% paraformaldehyde (PFA) for 30 min. PFA was then aspirated and cells were washed four times before quenching with 100 mM NH$_4$Cl (Merck, 12125-02-9) in PBS for 5 min. Cells were washed 3x with PBS followed by permeabilization with 0.2% Triton in PBS for 20 min. Afterwards, the sample was washed again 3x with PBS, and the blocking buffer was incubated for 45 min. After blocking, the sample was washed 3x with PBS, and gold nanoparticles (Cytodiagnostics) 1:2 diluted in PBS, were incubated for 5 min as fiducials. The sample was washed again 4x with PBS. Initially, primary antibody and secondary nanobody incubation of Golga5 and βII-Spectrin, pooled in 300 μl of antibody incubation buffer, was done onto the sample overnight at 4 °C, softly shaking on a belly dancer. Simultaneously, preincubation of the remaining antibodies with their respective secondary nanobodies was prepared according to Supplementary Tables 6,7, each in 10 μL antibody incubation buffer at 4 °C overnight while softly shaking on a belly dancer. On the next day, the sample was washed 4x with PBS and once with buffer C. Then an excess (molar ratio of 1:2) of the respective unlabeled secondary nanobodies (NanoTag Biotechnologies, cat: K0102-50) was introduced to the preincubated primary antibody to secondary nanobody pairs for 5 min. Subsequently, the pairs and PSD95 and Synaptotagmin nanobodies were pooled into 300 μL of antibody incubation buffer supplemented with 1:1000 unlabeled secondary nanobodies, and the mix was incubated with the sample for 70 min at RT, softly shaking on a belly dancer. After incubation, the sample was washed 4x with PBS and once with 2xSSC. For PSD95 and Synaptotagmin direct nanobodies, secondary labels (100 nM) were hybridized for 15 min in 500 μL optimized hybridization buffer (with BC13 and BC14 due to binder availability). The sample was washed 5x with 2xSSC and once with buffer C. The first imaging solution was introduced according to Supplementary Table 7. All targets were imaged with 16 mW laser power in HILO[20] illumination mode with 7.500 frames (75 ms frames), resulting in roughly 10 min acquisition time per protein target. Between targets, the sample was washed 4x with buffer C before the next imaging solution was introduced until all twelve targets were imaged. As a last target, Actin was imaged using Lifeact[12] with an imager solution according to Supplementary Table 7. The total experimental time is calculated using a single acquisition round of ~10 min (40 min for the entire 200 × 200 μm$^2$ FOV) and 2 min of buffer exchange, resulting in 544 min (or ~9.1 h) for all 13 targets. Achieving similar sampling and resolution with conventional Exchange-PAINT, based on comparison of $k_{on}$ values[4], would have taken ~11 h (~44 h for the entire 200 × 200 μm$^2$ FOV) per single target and in total for all 13 protein targets, nearly a month to complete. While the main benefit over SUM-PAINT and FLASH-PAINT is that secondary barcodes can be avoided, the combination of left-handed and right-handed multiplexing saves ~30 min in comparison to SUM-PAINT. For detailed comparison, see Supplementary Table 9. Multiplexed imaging rounds were drift-corrected and aligned with the help of gold fiducials (Cytodiagnostics G-90-100). For stitching, images were acquired with a 10 μm overlapping offset. Subsequently, the four individual FOVs were coarsely aligned manually based on the Tom20 channel within a few hundreds of nm in range. Overlapping areas where then registered with Picasso: Render and these regions were used for precise alignment through a custom script based on scipy.signal.correlate2d[25]. Using this alignment based on the overlapping regions, the top left FOV was used as a reference to which the bottom left and top right FOVs were aligned. The bottom right ROI was aligned to the top right FOV. Lastly, this precise alignment was transferred to all 13 channels and all channels were aligned accordingly.

## Reporting summary

Further information on research design is available in the Nature Portfolio Reporting Summary linked to this article.

## Data availability

Raw localization microscopy datasets (in hdf5 format) have been deposited at Zenodo and are publicly available as of the date of publication with this link: https://doi.org/10.5281/zenodo.17023207. Source data are provided with this paper.

## Code availability

All original code has been deposited at Zenodo and is publicly available as of the data of publication with this link: https://doi.org/10.5281/zenodo.17023207.

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

## Acknowledgments

This research was funded in part by the European Research Council through an ERC Consolidator Grant (ReceptorPAINT, Grant agreement number 101003275), the BMBF (Project IMAGINE, FKZ: 13N15990), the Max Planck Foundation, and the Max Planck Society. E.M.U., E.S., I.B., and T.T. acknowledge support from the IMPRS-ML graduate school. L.A.M. acknowledges a postdoctoral fellowship from the European Union's Horizon 2021-2022 research and innovation program under the Marie Skłodowska-Curie grant agreement no. 101065980. F.O. acknowledges support by Deutsche Forschungsgemeinschaft (DFG) through the SFB1286 (project Z04). E.F.F. is funded by a CZI collaborative pair grant. E.F.F. also acknowledges the support of the Collaborative Research Center 1286 on Quantitative Synaptologie (CRC/SFB1286), Göttingen, Germany.

## Author contributions

E.M.U., E.S., I.B., and T.T. shared experimental work, conceptualization, discussion, writing, and editing and contributed equally to the manuscript. J.K. conjugated and benchmarked labeling reagents. S.S. contributed to the conceptualization and performed initial experiments. K.J., F.O., and E.F.F. provided nanobodies and neuron samples and contributed to the design of the neuron multiplexing experiments. R.K. provided support with data analysis. L.A.M. and R.J. supervised the project and edited the final version of the manuscript. All authors reviewed and approved the final manuscript.

## Funding

## Competing interests

F.O. is a shareholder of NanoTag Biotechnologies GmbH. The other authors declare no competing financial interests.
