## [Transparent Peer Review file · Nature Communications]

Left-handed DNA for efficient highly multiplexed imaging at single-protein resolution

Corresponding Author: Professor Ralf Jungmann

Version 0:

Reviewer comments:

Reviewer #1

(Remarks to the Author)

In this manuscript, Unterauer et al. developed a streamlined 12-plex DNA-PAINT method, eliminating the intricate secondary labeling step used in their previous approach (Unterauer et al., 2024). By combining six orthogonally speed-optimized right-handed DNA sequences (Strauss et al., 2020) with their corresponding left-handed counterparts (Geertsema et al. 2021), the authors effectively doubled the repertoire of speed-optimized sequences, enabling 12-plex DNA-PAINT through a straightforward Exchange-PAINT workflow (Jungmann et al., 2014). Following their previous validation strategy, the method was first tested on DNA origami and subsequently applied to neuronal imaging. This is an interesting development that promises to be useful to the DNA-PAINT imaging community. The results are generally convincing, however several concerns should be addressed to improve clarity.

Major concerns:

Line 47: The authors claim a 100-fold imaging speed improvement compared to previous L-DNA-PAINT implementation (Geertsema et al. 2021). However, this claim lacks detailed quantification. It appears to be based on previous optimizations of buffer conditions (2-fold) and DNA sequences (5-fold) (Schueder et al., 2019), combined with the use of periodic motif design (up to ~10-fold, Strauss et al., 2020). Nevertheless, the optimized buffer conditions were not applied in either the DNA origami or the neuronal samples presented here. Additionally, most periodic motifs are unlikely to contribute a full 10-fold enhancement in speed. Although Lines 330-333 (method section) provide a general explanation for the claimed 100-fold improvement, it seems to result from a combination of shorter exposure times and reduced frame counts (75 ms and 4,500 frames in this study vs 100 ms and 10,000 frames in SUM-PAINT). Overall, the authors should be more transparent about how this was calculated, including a clearer breakdown and justification for this claim, and even better, present a direct comparison with SUM-PAINT in the main text rather than cryptic calculations buried solely in the methods section.

The DNA origami images reveal that in some cases there are missing sites in the images, suggesting variable labeling efficiency for the different docking oligos. It would be useful for the community if the authors quantified the labeling efficiency both from the DNA origami and the NUP images and report these values in the manuscript.

Figure 1d and 1e: The reported τ_d values in both the DNA-origami and NUP experiments exceed 100 seconds. However, in the authors' previous studies, the τ_d of R1 was reported as 13 seconds (Strauss et al., 2020) and approximately 50 seconds in its secondary labeling form (Unterauer et al., 2024), with the latter expected to shorten τ_d . These values are substantially shorter than those reported in the current study. Although the authors mention normalization based on imager concentration and the average number of binding sites in the methods section, this information is not reflected in the figures. I encourage the authors to clarify this discrepancy and provide a more appropriate evaluation if possible.

Line 67-68: The authors state that the left-handed DNA sequences exhibit similar speed-optimized kinetics to their right-handed counterparts. However, several of the paired sequences used in the DNA origami experiments (for example, sequences 1, 2, 5, and 6) appear to show noticeable differences, as clearly illustrated in the supplementary figures. Additionally, Geertsema et al. reported highly similar binding kinetics between mirrored sequences such as P3 and LP3. These observations raise the question of why such kinetic differences arise, given prior reports of comparable behavior between mirrored sequences. Could the authors elaborate on this apparent inconsistency? In addition, the authors should reword their claim since in some cases the kinetics differ substantially.

Line 62-64: Some super-resolved biological features, such as the 12 nm spacing between Nup96 copies and the hollowness of the microtubule, are described but not quantitatively analyzed or visually represented. It would strengthen the manuscript if the authors could provide the corresponding quantifications and show examples of microtubules where the hollow lumen is visible.

The authors present a demonstration of 3D imaging over a $200 \times 200 \mu\text{m}^2$ field of view by stitching four FOVs. However, the Methods section does not provide sufficient details on the 3D SMLM imaging strategy, or the procedure used for FOV alignment, particularly in 3D. Providing additional information on these aspects would be helpful for readers to better understand and evaluate the approach.

Other comments:

Figure 2b: it's very hard to see these panels as the contrast is very low.

Figure 2d: Could the authors specify where the images in the right panels are located within the left panel?

Figure 1c: There is a typo in the label "Microtubuli".

Line 249-250 & Supplementary Table 8: There is a mismatch in the reported laser power. The methods section states 30 mW, while the table lists 50 mW.

Line 259 & Supplementary Table 8: There is a discrepancy in the number of frames reported. The methods section states 80,000 frames, whereas the table indicates 100,000 frames.

Supplementary Table 8: In the fifth row, second column, "Supplementary Table 7" should be corrected to "Supplementary Table 6".

Reviewer #2

(Remarks to the Author)

Reviewer #3

(Remarks to the Author)

In the manuscript titled "Left-handed DNA for efficient highly multiplexed imaging at single-protein resolution", Unterauer et al. have introduced a fast and sequence-optimized DNA-PAINT imaging based on Left-handed of DNA helices, developing upon the previous work of Geertsema et al., 2021 and their own series of previous work described in the manuscript. Overall, the manuscript is well-written and lucidly explained making it easy for the reader to understand. I am mostly satisfied with the work, provided that the authors address my concerns. My concerns and suggestions are listed below:

1. In supplementary 1, in addition to the L-sequences, I would like to see the performance of R sequences at 5nm DNA-origami structure, under exact same imaging conditions and understand how poorly they perform.
2. What are the persistent lengths of these R and L sequences? Are they comparable to naturally occurring B and Z DNA? The naturally occurring left-handed DNA sequences, also called the Z-DNA, has a persistence lengths have significantly reduced persistence length. Has the rigidity or the flexibility of these DNA sequences tested. Even though intuitively, chirality should not impact the rigidity, but the changes in the overall structure can.
3. The L-binders seem to have starkly different On times (almost 2-4.5 fold difference) with respect to each other (for example, L1, 2, 3 has considerably lower on time compared to L2, 4, 6), considering the variability in the binding times of each binder. This inter-imager variability in On time seems to have reduced in the case of Nuclear pore imaging. Can the authors explain, what might have lead to such inter-imager variability in the DNA origami but not in nuclear pore? Would the on and off times be different for the 5nm vs 15nm Origami?
4. This gives rise to my following question: If these L-sequences used for detecting various proteins within a specimen, would the differences in binding time not impact the number of detected localizations between proteins? In that case, would the method be useful to obtain an absolute copy numbers, variance and the inter-relationships of various proteins in the cell?
5. "After validation in synthetic nanostructures, we evaluated the performance of the L-DNA strands in cellular environments"- the cell type and culture condition must be mentioned. What is the effective labeling efficiency of the benchmarking structures? Do these sequences have over or under labeling bias? Because that could be detrimental for molecular counting. Why different imaging frame numbers were used in this experiment while for the other experiments the total imaging frame number remains constant?
6. No information about the antibodies used were given in the methods section, making it hard to understand the nature of these binders and how they may influence the morphology of the structures. Given the similar method has been used in distance measurement in one of their previous publication and therefore, is a potential application of the current development as well, it is necessary that the readers have an access to this information. For example, proteins like Basoon has been reported to span lengths scales of 100s of nanometes. And if the authors used monoclonal antibodies or

nanobodies, they may have ended up capturing only a part of the entire proteins length span, thus, may significantly impact their distance calculations from other neighboring proteins. What do the authors think is a better way to represent such a lengthy structures in terms of localization? Certainly, coordinate points of the center of mass seems to be an oversimplification of the problem.

Overall, my major concern is the differences in their bright time which is a function of both the imager binding time and the stochastic bleaching of the blinking fluorophores.

Reviewer #4

(Remarks to the Author)

The authors combine speed-optimized right-handed and left-handed DNA-PAINT sequences to enable efficient, highly multiplexed imaging at single-protein resolution. The particularly exciting thing about this manuscript is that the approach drastically reduces imaging time without compromising the resolution to improve the major weakness of traditional super-resolution imaging based on DNA-PAINT. Especially, the time cost to map a $200 \times 200 \mu\text{m}^2$ field, including 12 targets, is reduced from traditionally up to a month to within 10 hours. This approach implements the simultaneous investigation of molecular and network-level features. Personally, the research design is sound, and the proposed ideas and results are potentially attractive to a wide audience for imaging and precise therapy.

Here are a few specific questions for clarification:

1. The manuscript title "Left-handed DNA for efficient highly multiplexed imaging at single-protein resolution" might confuse the audience that the approach and corresponding results are completed without right-handed DNA. Maybe "Combining left-handed DNA for efficient highly multiplexed imaging at single-protein resolution" or another revision?
2. In the manuscript, the authors state, "We note that our approach can be combined with approaches like peptide-PAINT (in this case Lifeact for a 13-plex image), further expanding the multiplexing". Since DNA-PAINT with orthogonal sequence design enables theoretically unlimited multiplexing, the reason and necessity for expanding multiplexing by another method should be described in more detail.
3. A representative characterization result of a nanobody-DNA conjugation is encouraged to be supplemented in supplementary information, such as that obtained by PAGE or AGE.

Version 1:

Reviewer comments:

Reviewer #1

(Remarks to the Author)

The authors have addressed all our comments and we have no further comments.

Reviewer #2

(Remarks to the Author)

Reviewer #3

(Remarks to the Author)

The Authors have fully addressed my concerns.

Reviewer #4

(Remarks to the Author)

As the authors have answered all the questions properly, I recommend accepting it for publication without further comments.

REVIEWER COMMENTS

Reviewer #1 (Remarks to the Author):

In this manuscript, Unterauer et al. developed a streamlined 12-plex DNA-PAINT method, eliminating the intricate secondary labeling step used in their previous approach (Unterauer et al., 2024). By combining six orthogonally speed-optimized right-handed DNA sequences (Strauss et al., 2020) with their corresponding left-handed counterparts (Geertsema et al. 2021), the authors effectively doubled the repertoire of speed-optimized sequences, enabling 12-plex DNA-PAINT through a straightforward Exchange-PAINT workflow (Jungmann et al., 2014). Following their previous validation strategy, the method was first tested on DNA origami and subsequently applied to neuronal imaging. This is an interesting development that promises to be useful to the DNA-PAINT imaging community. The results are generally convincing, however several concerns should be addressed to improve clarity.

Major concerns:

Line 47: The authors claim a 100-fold imaging speed improvement compared to previous L-DNA-PAINT implementation (Geertsema et al. 2021). However, this claim lacks detailed quantification. It appears to be based on previous optimizations of buffer conditions (2-fold) and DNA sequences (5-fold) (Schueder et al., 2019), combined with the use of periodic motif design (up to ~10-fold, Strauss et al., 2020). Nevertheless, the optimized buffer conditions were not applied in either the DNA origami or the neuronal samples presented here. Additionally, most periodic motifs are unlikely to contribute a full 10-fold enhancement in speed. Although Lines 330-333 (method section) provide a general explanation for the claimed 100-fold improvement, it seems to result from a combination of shorter exposure times and reduced frame counts (75 ms and 4,500 frames in this study vs 100 ms and 10,000 frames in SUM-PAINT). Overall, the authors should be more transparent about how this was calculated, including a clearer breakdown and justification for this claim, and even better, present a direct comparison with SUM-PAINT in the main text rather than cryptic calculations buried solely in the methods section.

We thank the reviewer for raising this point and apologize for the confusion. The claim of a 100-fold imaging speed improvement was solely based on the extrapolation of the kinetics of Geertsema et al. 2021 (P-sequence kinetics with 1x speed) to Strauss et al. 2020 (Theoretically up to 100x speed with speed-improved R-sequences). Since, as the reviewer pointed out, we do not use the 10x speed-optimized sequences but the 5x speed-optimized sequences from Strauss et al. 2020, we removed the claim of a 100-fold speed improvement. Furthermore, to present a more comparable speed comparison between different methods, we calculated the k_{on} for each sequence and added the values in the now separated Supplementary Tables 4 and 5. The direct comparison of P1 imaging kinetics, $k_{on} = 1.25 * 10^6 (Ms)^{-1}$ (Schueder et al. Nature Methods 2019) to the kinetics calculated from our experiments with left-handed DNA-sequences, $k_{on} = 81.2 * 10^6 (Ms)^{-1}$, shows a 65-fold improvement. Based on these values and taking reported values from Strauss et al. (Nature Methods 2020), Schueder et al. (Cell 2024), and Unterauer et al. (Cell 2024) into account, we assembled the new Supplementary Table 9, which in addition to theoretical calculations summarizes the most important parameters (k_{on} , secondary-label usage, number of targets achieved, and the resulting times for single experiments and the 13-

plex neuronal atlas experiment in this paper) for an acquisition workflow and time comparison. We note that neither the shorter integration time of 75 ms nor the increases in imaging FoV contributed to the speed comparison.

The DNA origami images reveal that in some cases there are missing sites in the images, suggesting variable labeling efficiency for the different docking oligos. It would be useful for the community if the authors quantified the labeling efficiency both from the DNA origami and the NUP images and report these values in the manuscript.

We thank the reviewer for this comment. For DNA origami and NPC experiments, this is actually to be expected. For the DNA origami case, missing sites are due to variations in incorporation efficiency, characterized by Strauss et al. (Nature Communications 2018, doi:[10.1038/s41467-018-04031-z](https://doi.org/10.1038/s41467-018-04031-z)) and Unterauer et al. (Cell 2024, doi:[10.1016/j.cell.2024.02.045](https://doi.org/10.1016/j.cell.2024.02.045)). Regarding the labeling efficiency of NUP images, this labeling efficiency has also been characterized in detail by Thevathasan et al. (Nature Methods 2019, doi:[10.1038/s41592-019-0574-9](https://doi.org/10.1038/s41592-019-0574-9)) and Hellmeier et al. (Nature Methods 2024, doi:[10.1038/s41592-024-02242-5](https://doi.org/10.1038/s41592-024-02242-5)). We have therefore included references to these studies focusing on the labeling efficiency of Nup96 in the nuclear pores.

Figure 1d and 1e: The reported τ_d values in both the DNA-origami and NUP experiments exceed 100 seconds. However, in the authors' previous studies, the τ_d of R1 was reported as 13 seconds (Strauss et al., 2020) and approximately 50 seconds in its secondary labeling form (Unterauer et al., 2024), with the latter expected to shorten τ_d . These values are substantially shorter than those reported in the current study. Although the authors mention normalization based on imager concentration and the average number of binding sites in the methods section, this information is not reflected in the figures. I encourage the authors to clarify this discrepancy and provide a more appropriate evaluation if possible.

We thank the reviewer for pointing out this difference in reported τ_d values. As different imager concentrations have been used for these, differences in the τ_d values are expected (according to $\tau_d = 1/(k_{on} * c)$). In order to report an independent association kinetic from the imager concentration, we have now calculated the k_{on} values for all sequences to enable a comparison of kinetics independent of the imager concentration. The k_{on} values were combined in the now separated Supplementary Tables 4 and 5. The comparison of k_{on} to prior studies was conducted in the new Supplementary Table 9.

Line 67-68: The authors state that the left-handed DNA sequences exhibit similar speed-optimized kinetics to their right-handed counterparts. However, several of the paired sequences used in the DNA origami experiments (for example, sequences 1, 2, 5, and 6) appear to show noticeable differences, as clearly illustrated in the supplementary figures. Additionally, Geertsema et al. reported highly similar binding kinetics between mirrored sequences such as P3 and LP3. These observations raise the question of why such kinetic differences arise, given prior reports of comparable behavior between mirrored sequences. Could the authors elaborate on

this apparent inconsistency? In addition, the authors should reword their claim since in some cases the kinetics differ substantially.

We thank the reviewers for pointing out this important topic. While the left-handed DNA-PAINT sequences show similar kinetics in terms of speed-improvement such as from P-sequences to R-sequences, they do differ from each other and from left-handed to right-handed geometry. To avoid confusion, we have removed this claim and added a detailed discussion of the kinetic differences in the main text, including possible explanations based on the base stacking interaction differences of the left-handed and right-handed sequences. While Geertsema et al. reported similar kinetics between right-handed and left-handed sequences, we also observe this similarity for e.g. left-handed/right-handed L3/R3 and L4/R4 as the reviewers pointed out. Since Geertsema et al. only analyzed the kinetics of P3 and its left-handed complement, these differences could appear between different P-sequences.

Line 62-64: Some super-resolved biological features, such as the 12 nm spacing between Nup96 copies and the hollowness of the microtubule, are described but not quantitatively analyzed or visually represented. It would strengthen the manuscript if the authors could provide the corresponding quantifications and show examples of microtubules where the hollow lumen is visible.

We thank the reviewers for this suggestion and have included a distance histogram of the Nup96 copies (spaced 13.7 nm apart). We have included a cross-section of a microtubule, but have removed our claim of resolving the hollowness of the microtubule structure, as it is not completely resolvable in these measurements.

The authors present a demonstration of 3D imaging over a $200 \times 200 \mu\text{m}^2$ field of view by stitching four FOVs. However, the Methods section does not provide sufficient details on the 3D SMLM imaging strategy, or the procedure used for FOV alignment, particularly in 3D. Providing additional information on these aspects would be helpful for readers to better understand and evaluate the approach.

We thank the reviewer for pointing this out. We have now added the information describing our 3D astigmatism based imaging approach and the overlay of the 13-plex neuronal atlas based on overlapped FOVs and neurofilament structure in the methods section.

Other comments:

Figure 2b: it's very hard to see these panels as the contrast is very low.

We thank the reviewers for pointing this out and have increased the contrast of the panels in Figure 2b.

Figure 2d: Could the authors specify where the images in the right panels are located within the left panel?

We thank the reviewers for this important note. The selection of zoom-ins on the right panels comes from different regions than the left panel. We have added additional descriptions in the figure caption to avoid this confusion.

Figure 1c: There is a typo in the label “Microtubuli”.

We thank the reviewers for pointing this out and have now corrected the typo.

Line 249-250 & Supplementary Table 8: There is a mismatch in the reported laser power. The methods section states 30 mW, while the table lists 50 mW.

We thank the reviewers for highlighting this point and have now corrected the mismatch.

Line 259 & Supplementary Table 8: There is a discrepancy in the number of frames reported. The methods section states 80,000 frames, whereas the table indicates 100,000 frames.

We thank the reviewers for pointing this out and have corrected the discrepancy.

Supplementary Table 8: In the fifth row, second column, “Supplementary Table 7” should be corrected to “Supplementary Table 6”.

We thank the reviewers for pointing this out and have corrected all Supplementary Table references.

Reviewer #2 (Remarks to the Author):

Reviewer #3 (Remarks to the Author):

In the manuscript titled “ Left-handed DNA for efficient highly multiplexed imaging at single-protein resolution”, Unterauer et. al. have introduced a fast and sequence-optimized DNA-PAINT imaging based on Left-handed of DNA helices, developing upon the previous work of Geertsema et. al., 2021 and their own series of previous work described in the manuscript. Overall, the manuscript is well-written and lucidly explained making it easy for the reader to understand. I am mostly satisfied with the work, provided that the authors address my concerns. My concerns and suggestions are listed below:

1. In supplementary 1, in addition to the L-sequences, I would like to see the performance of R sequences at 5nm DNA-origami structure, under exact same imaging conditions and understand how poorly they perform.

We thank the reviewer for this suggestion and have changed Supplementary Figure 1 to contain now both, an example of L1-Cy3B imaging of a 5 nm DNA origami and a R3-Cy3B DNA origami under exactly the same imaging conditions for comparability.

2. What are the persistent lengths of these R and L sequences? Are they comparable to naturally occurring B and Z DNA? The naturally occurring left-handed DNA sequences, also called the Z-DNA, has a persistence lengths have significantly reduced persistence length. Has the rigidity or the flexibility of these DNA sequences tested. Even though intuitively, chirality should not impact the rigidity, but the changes in the overall structure can.

We thank the reviewer for raising this point. We have not directly quantified the persistence lengths of these specific R and L sequences, but we believe that they should be similar to naturally occurring B and Z DNA. We expect synthetic left-handed DNA sequences to also have a somewhat reduced persistence length. However a detailed study of the physicochemical parameters of the L and R sequences would require further dedicated studies, beyond the scope of this work.

3. The L-binders seem to have starkly different On times (almost 2-4.5 fold difference) with respect to each other (for example, L1, 2, 3 has considerably lower on time compared to L2, 4, 6), considering the variability in the binding times of each binder. This inter-imager variability in On time seems to have reduced in the case of Nuclear pore imaging. Can the authors explain, what might have lead to such inter-imager variability in the DNA origami but not in nuclear pore? Would the on and off times be different for the 5nm vs 15nm Origami?

We thank the reviewer for raising this point. The difference in ON time originates from the variation of base interactions of the sequences. With limited design options (a 3-letter base code), the binding times can vary starkly (as pointed out by the reviewer), which is in fact similar to the previous implementation of right-handed speed-optimized sequences by Strauss et al. (Nature Methods 2020).

With regards to the variability between R- and L-DNA kinetics as well as DNA origami vs. NPC experiments, we have now added a clarifying section in the main text, that reads as follows:

“However, there are certain variations in binding kinetics between DNA origami and NPC experiments as well as between left- and right-handed sequences. The differences in the former case could potentially be attributed to differences in experimental settings: DNA origami strand extensions are hybrid sequences transitioning from right-handed to left-handed motifs in a single strand, which possibly alters the base-stacking interactions in the vicinity of the transition. The differences in the latter case, specifically the bright times of R2 and L2 differ in both the DNA origami and NPC contexts, a phenomenon for which we currently lack an explanation.”

Finally, given the same experimental conditions, we do not expect any considerable differences between the on- and off-times in 5 vs 15 nm spaced DNA origami structures.

4. This gives rise to my following question: If these L-sequences used for detecting various proteins within a specimen, would the differences in binding time not impact the number of detected localizations between proteins? In that case, would the method be useful to obtain an absolute copy numbers, variance and the inter-relationships of various proteins in the cell?

We thank the reviewer for their comment. Only differences in inter-event lifetimes (also called dark times or τ_d) would impact counting, as differences in ON-time (or bound/bright time) are taken into account when grouping localizations to transition from localizations to actual binding events. When conducting a quantitative DNA-PAINT experiment (qPAINT, Jungmann et al Nature Methods 2016), a dark-time measurement of a calibration structure with a known number of binding sites is needed to obtain high counting accuracy in qPAINT. Once calibrated, an absolute estimate of the number of molecules can be obtained for a given species of molecules or protein targets. If the task is to compare molecules of different species with independent imager/docking strand pairs, the same calibration needs to be done again for each sequence used.

5. “After validation in synthetic nanostructures, we evaluated the performance of the L-DNA strands in cellular environments”- the cell type and culture condition must be mentioned.

We thank the reviewer for pointing this out and have now added the cell type in the main part of the manuscript. Details about cell culture can be found in the Methods section (“3-plex imaging in U2OS cells.”)

What is the effective labeling efficiency of the benchmarking structures? Do these sequences have over or under labeling bias? Because that could be detrimental for molecular counting.

We thank the reviewer for pointing us to this topic. All labeling reagents have an “under-labeling” bias because no affinity reagent can reach 100% labeling efficiency. In case of the nuclear pore protein Nup96 tagged with a GFP and labeled by a GFP nanobody, this labeling efficiency has been thoroughly characterized by Hellmeier et al. (Nature Methods 2024) and Thevathasan et al. (Nature Methods 2019) to be approximately 50%. For the other benchmarking targets (Tom20 with a rabbit secondary nanobody and alpha-tubulin with a mouse secondary nanobody), no labeling efficiencies have been reported. While we make no claim about molecular

counting of these structures, the reviewer is correct that if molecular counting, referring to the ground truth biological structure, was the aim of the study, all labeling reagents would have to be characterized for labeling efficiency. The sequences themselves do not contribute to over- or under-counting artifacts as described in the response to question #4.

Why different imaging frame numbers were used in this experiment while for the other experiments the total imaging frame number remains constant?

For the benchmarking structure imaging in the U2OS cells, the imaging concentration was adjusted for optimal sampling of the three target structures (Nup96, Tom20 and alpha-tubulin). Since these structures have different labeling densities, the imager concentration (and hence blinking frequency) was adjusted for optimal resolution ensuring non-overlapping single-molecule signals from imager strands. On the other hand, for the kinetics comparison experiments as well as for the multiplexed 12-plex DNA origami and 13-plex neuronal atlas, the focus was not on optimal resolution or sampling of the target, but rather on the overall imaging workflow, which we aimed to keep simple and robust. If needed, the sampling duration can be adjusted and fine-tuned for every round to achieve optimal experimental resolution and sampling.

6. No information about the antibodies used were given in the methods section, making it hard to understand the nature of these binders and how they may influence the morphology of the structures.

We thank the reviewer for this important note and apologies for this oversight. The antibodies used in this study are all compiled in Supplementary Table 6, with full information about dilution and vendors. We have now also added references to this table in the main text (which were missing by mistake). The incubation duration is described in the methods section for each experiment individually.

Given the similar method has been used in distance measurement in one of their previous publication and therefore, is a potential application of the current development as well, it is necessary that the readers have an access to this information. For example, proteins like Basoon has been reported to span lengths scales of 100s of nanometres. And if the authors used monoclonal antibodies or nanobodies, they may have ended up capturing only a part of the entire proteins length span, thus, may significantly impact their distance calculations from other neighboring proteins. What do the authors think is a better way to represent such a lengthy structures in terms of localization? Certainly, coordinate points of the center of mass seems to be an oversimplification of the problem.

We thank the reviewer for raising this point. As for any immunolabeling-based microscopy method, we are limited by the accuracy provided by the labels that are available. Labeling molecules will introduce a bias to the measurements (the smaller the molecule or molecular complex the smaller the bias). A novel way to represent localizations taking into account the size of the label used is for sure an interesting idea that would be worth exploring. For this work we decided to use the standard and most commonly used representations of

localizations in the field of super-resolution. This, together with the detailed information about the labeling molecules (antibodies) should provide the reader the information needed to interpret the data correctly, including its limitations. We further note that the same antibody reagents were used as for our earlier study (Unterauer et al., Cell 2024).

Overall, my major concern is the differences in their bright time which is a function of both the imager binding time and the stochastic bleaching of the blinking fluorophores.

We thank the reviewer for raising this concern. While the difference in bright times for the left-handed DNA-PAIN sequences is a factor that needs to be taken into account when choosing the respective imager concentration, this difference is to be expected due to sequence difference, reported by Strauss et al. (Nature Methods 2020) for right-handed speed sequences. Regarding the stochastic bleaching of the blinking fluorophores, we note that we work at laser powers that are far from the bleaching point of the fluorophores (which in our case are very photostable as the blinking is only governed by the DNA hybridization kinetics and not by any photophysics).

Reviewer #4 (Remarks to the Author):

The authors combine speed-optimized right-handed and left-handed DNA-PAINT sequences to enable efficient, highly multiplexed imaging at single-protein resolution. The particularly exciting thing about this manuscript is that the approach drastically reduces imaging time without compromising the resolution to improve the major weakness of traditional super-resolution imaging based on DNA-PAINT. Especially, the time cost to map a $200 \times 200 \mu\text{m}^2$ field, including 12 targets, is reduced from traditionally up to a month to within 10 hours. This approach implements the simultaneous investigation of molecular and network-level features. Personally, the research design is sound, and the proposed ideas and results are potentially attractive to a wide audience for imaging and precise therapy.

Here are a few specific questions for clarification:

1. The manuscript title “Left-handed DNA for efficient highly multiplexed imaging at single-protein resolution” might confuse the audience that the approach and corresponding results are completed without right-handed DNA. Maybe “Combining left-handed DNA for efficient highly multiplexed imaging at single-protein resolution” or another revision?

We thank the reviewer for bringing up this point. We understand the potential concern that the current title, “*Left-handed DNA for efficient highly multiplexed imaging at single-protein resolution*,” might be interpreted as implying that the approach is performed exclusively with left-handed DNA.

However, we would like to clarify that the title was chosen to highlight the key point in our study, namely the use of left-handed DNA as the central enabling element for efficient, highly multiplexed single-protein imaging. Therefore, we would prefer to retain the original title, as it most directly conveys the central message and of our study without introducing additional complexity.

2. In the manuscript, the authors state, “We note that our approach can be combined with approaches like peptide-PAINT (in this case Lifeact for a 13-plex image), further expanding the multiplexing”. Since DNA-PAINT with orthogonal sequence design enables theoretically unlimited multiplexing, the reason and necessity for expanding multiplexing by another method should be described in more detail.

We thank the reviewer for raising this point. For certain protein targets, DNA-conjugated labeling reagents such as antibodies or nanobodies are not the preferred option. Point in case is PAINT-based actin imaging, as using a small, transiently binding, dye-labeled peptide motif (LifeAct, which also was optimized by us for PAINT-like super-resolution imaging) is in fact superior to protein-based binders such as antibodies.

3. A representative characterization result of a nanobody-DNA conjugation is encouraged to be supplemented in supplementary information, such as that obtained by PAGE or AGE.

We thank the reviewer for this suggestion and have added Supplementary Figure 8, showing three separate anion exchange chromatograms of DNA, unconjugated DNA and DNA-conjugated GFP nanobody.